# Clinical Characteristics of Venous Thrombosis Associated with Peripherally Inserted Central Venous Catheter in Premature Infants

**DOI:** 10.3390/children9081126

**Published:** 2022-07-28

**Authors:** Weiwei Zhu, Hua Zhang, Yan Xing

**Affiliations:** 1Pediatric Department, Peking University Third Hospital, Beijing 100191, China; yeshangwuzhebang@163.com; 2Clinical Epidemiology Research Center, Peking University Third Hospital, Beijing 100191, China; zhanghua824@163.com

**Keywords:** preterm infants, autoimmune disease, venous thrombosis, anticoagulation therapy

## Abstract

Background: This study aimed to analyze clinical characteristics and risk factors for peripherally inserted central catheter (PICC) placement in premature infants. Materials: This study included seven premature infants who were hospitalized in the neonatal intensive care unit (NICU) of Peking University Third Hospital from 1 January 2014–30 June 2021, and suffered PICC-related venous thrombosis. The control group included premature infants (*n* = 56) matched (1:8) by the following: Did not experience venous thrombosis; born with a similar gestational age (±2 w), birth weight (±200 g); and received PICC catheterization in the same period (±4 w). Clinical neonatal data were collected through the hospital electronic medical record system and analyzed using SPSS version 23. Results: The incidence of PICC-related thrombus was 0.23% (7/3043. Univariate analysis revealed that, compared to the non-thrombotic group, mothers in the thrombosis group had autoimmune diseases (χ^2^ = 9.844, *p* = 0.030) and used anticoagulative drugs during pregnancy (χ^2^ = 8.036, *p* = 0.025). The corrected gestational age when PICC-related thrombosis occurred in the thrombosis group was 32 + 6 (30 + 1, 34 + 1) weeks. The average time from catheter placement to thrombosis was 5 (1, 12) days. Among infants, 85.7% (6/7) experienced deep vein thrombosis, of which four were in the lower extremity veins; three occurred within 2 days after central venous catheter extubation, and four occurred during central venous catheter indwelling. The clinical manifestations of thrombosis include skin edema, color changes, and skin temperature changes in the affected limbs. The seven neonates had normal coagulation at the time of thrombus diagnosis, but D-dimers significantly increased 1–2 days after thrombosis, returning to normal 5–8 days after thrombus. The thrombus persisted for 4.5 (3, 8) days. All seven neonates were treated with low molecular weight heparin calcium anticoagulation for 10 (3, 17) days and recovered completely. Conclusions: PICC-related thrombosis occurred within 1 week after catheter placement, and thrombosis more likely happened in infants whose mothers had autoimmune disease. When this high-risk factor exists and the patient has been intubated for 1 week and has sudden swelling in the intubated limb, venous ultrasound should be performed immediately to diagnose, and treatment should be provided in a timely manner to reduce adverse events.

## 1. Introduction

Peripherally inserted central catheters (PICC) are commonly used in neonatal intensive care units (NICUs) to treat critically ill neonates, especially premature infants. However, PICC can cause complications such as catheter-related blood stream infection (CRBSI), chylothorax, chyloma, arrhythmia, catheter displacement, phlebitis, and venous thrombosis [1,2,3,4]. PICC-related venous thrombosis is one of the more serious complications. However, there are few reports on PICC-related venous thrombosis, and there is no consensus on thrombosis risk factors, treatments, and prognosis. This study aimed to analyze the clinical characteristics of PICC-related venous thrombosis and determine potential risk factors, treatment efficacy, and prognosis. Assessing these aspects of patient care should provide evidence for clinical prevention, early identification, and effective treatment of PICC-related venous thrombosis.

## 2. Materials and Methods

### 2.1. Participants and Groups

This retrospective cohort study included 63 premature neonates who were admitted to the Neonatal Intensive Care Unit at Peking University Third Hospital after birth and received PICC during hospitalization from 2014–June 2021. A total of 3043 cases of PICC catheterization were performed during this period. Patients were excluded if they died or were transferred to pediatric surgery due to critical conditions during PICC indwelling. The participants were classified into two groups based on whether they experienced PICC-related venous thrombosis (*n* = 7) or did not develop venous thromboses (*n* = 56, control). All participants were born with similar gestational age (±2 w) and birth weight (±200 g); they were hospitalized and received PICC catheterization during the same period (±4 w). Peking University Third Hospital Ethics Committee provided ethical approval for this study.

The diagnosis of PICC-related venous thrombosis was based on the occurrence of thrombosis in the vein where the catheter was inserted during the indwelling period or within 3 days after removal of the peripherally inserted central venous catheter and confirmed by vascular ultrasound.

### 2.2. Clinical Presentation

The medical records of the neonates and their mothers were retrieved through the hospital electronic medical record system. The clinical data included patient gestational age, birth weight, gender, maternal comorbidities, and medication history. PICC-related data included time of catheter insertion, operator, vein selection, catheter tip placement, and time from cannulation to thrombosis. Additionally, data were collected for clinical information before PICC (diagnosis, medication, platelet level), clinical information after PICC (coagulation function, platelet level, anticoagulant drug side effects), thrombosis management, and patient prognosis.

### 2.3. Statistical Analysis

All analyses were performed using SPSS version 23 (IBM, New York, NY, USA). The Kruskal–Wallis test was used for continuous variables. Significance was set to a *p*-value < 0.05. For continuous variables conforming to a normal distribution, we calculated mean (±standard deviation) for data conforming to the normal distribution; and as median (Q1, Q3) for the non-normally distributed data. For categorical variables, we calculated frequency counts and percentages against the total number of participants or event-specific participants. Continuous variables were evaluated using Student’s *t* test or Mann–Whitney rank-sum test as appropriate. Categorical variables were assessed through the χ^2^ or the Fisher exact test, as appropriate. Univariate and multivariate logistic regression analyses were conducted to calculate the unadjusted and adjusted odds ratio (OR) and 95% confidence interval (CI) for the association between the exploratory variables and PICC-related thrombosis.

## 3. Results

### 3.1. PICC-Related Thromboses

The incidence of PICC-related thrombosis was 0.23% (7/3043). There were no significant differences in gestational age, birth weight, and gender between the thrombosis group and the non-thrombotic group (*p* > 0.05). The data are listed in Table 1.

### 3.2. Risk Factors for PICC-Related Venous Thrombosis

Table 2 lists several risk factors for PICC-related venous thrombosis. The results of univariate analysis showed that, compared with the non-thrombotic group, there were statistically significant differences in maternal and gestational autoimmune diseases and use of anticoagulants (*p* < 0.05).The multivariate analysis showed that mothers’ use of anticoagulants, patients with cardiac insufficiency, or being TTTS-donor may have increased the risk of thrombosis.

### 3.3. Clinical Features of PICC-Related Thrombosis

Table 3 presents the clinical characteristics, treatments, and prognoses of seven neonates with PICC-related thrombosis. In four cases, the PICCs were placed in the great saphenous vein (two cases each on the left and right sides), while the remaining three cases were on the Guiyao veins (two cases on the right side, one case on the left side). The thrombus occurred in the deep vein in six cases, and in one case in the superficial vein. The average time from catheter placement to thrombosis formation was 5 (1, 12) days. The case 3 thrombosis occurred 36 days after catheterization. This patient was diagnosed with Staphylococcus aureus septicemia before thrombosis. The culture of the PICC tube confirmed the same bacteria, and it was considered that the thrombosis in this patient was related to the local bacterial retention in the catheter. Among the seven neonates, thrombosis clinically manifested as swelling in the involved limbs and related areas; three patients experienced changes in the color of the skin, one patient experienced increased skin temperature, and one patient had induration and tenderness. The platelet count within 24 h of thrombosis formation was 273 (57, 440). The coagulation function test was completed on the day when the thrombus was found in six cases, and four cases were normal. D-dimer levels were 3 times higher than the upper limit of normal in two cases (1.06 µg/mL and 12.54 µg/mL, respectively). Among the four patients with normal D-dimer, the values increased in three patients after reexamination (from the 1st to 2nd day after thrombosis). The elevated D-dimer values all returned to normal in the later period. The protein C (%), protein S (%), and AT-III (%) values for five neonates were normal. When the thrombus occurred, one patient had the PICC removed immediately (this case was a superficial vein thrombus), in three cases it was not removed at the time, and in 3 cases, thrombus occurred after extubation. 

### 3.4. Treatment and Complications of Anticoagulation

All seven neonates were treated with elevation of the affected limb and subcutaneous injection of anticoagulants. This study used the nadroparin calcium injection (Superblin) as anticoagulant therapy. No thrombolytic therapy was performed. The absence of contraindications was verified by performing routine blood test, coagulation function, and head ultrasound examination prior to beginning anticoagulant therapy. The average course of low molecular weight heparin was 10 (3, 17) days. The average complete resolution was 4.5 (3, 8) days. 

One neonate used anticoagulation for 2 days and stopped due to intracranial hemorrhage (Grade II). Bleeding gradually stabilized after drug discontinuation.

The structure and function of all neonates’ affected limbs returned to normal (Table 4).

## 4. Discussion

The incidence of PICC-related thrombosis among the study participants was 0.23%. The incidence of neonatal PICC-related thrombosis varies widely among reported studies. The incidence of symptomatic PICC-related upper extremity deep vein thrombosis in neonates is 0.3–28.3%, and the incidence of superficial vein thrombosis is 3 times that of deep vein thrombosis [5]. Aditya et al. showed that the prevalence of thrombosis in PICCs is low, about 1%, and may be related to the use of ultrasound guidance in about two-thirds of PICC placements in this study [6].

This study found that neonates whose mothers had autoimmune diseases were more prone to developing thrombosis after PICC insertion. The mother of one neonate in the thrombosis group in our study had antiphospholipid syndrome (APS). Another mother had undifferentiated connective tissue disease (UCTD). Antiphospholipid syndrome is a non-inflammatory autoimmune mediated coagulation disorder, characterized by arteriovenous thrombosis, pathological pregnancy, and thrombocytopenia. Cardiolipin antibody has been reported in newborns born to mothers with APS, indicating that the antibody can pass through the placenta [7]. There are studies showing that thrombotic events occurred in newborns born to mothers with APS. Sixty percent of infants have at least one additional risk factor for thrombosis. The most common risks include arterial or venous catheterization, sepsis, asphyxia, and/or congenital thrombosis. This supports a hypothesis called “double whammy”. APS itself is often not enough to cause thrombosis but works in synergy with other risk factors [8]. One study followed 134 newborns born to mothers with APS for 5 years. Most APS-related antibodies from mothers are cleared within 6 months after birth, but some antibodies last up to 24 months. There were no cases of thrombosis in these children [9]. Undifferentiated connective tissue disease refers to having more than one clinical symptom or sign of connective tissue disease, accompanied by more than one positive autoantibody. However, it does not meet the diagnostic and classification criteria for any established connective tissue disease. There are few studies on the prognosis of neonates born to mothers with UCTD. Pregnant women with UCTD diagnosed in the first trimester may have a higher risk of fetal growth restriction or preeclampsia, but the risk of thrombosis has not been well studied in this patient population [10]. Currently, there are few studies on the influence of pregnant women with autoimmune diseases on the coagulation function of newborns, and further research is still needed to clarify the relationship between them.

We found that patients with cardiac insufficiency or being a TTTS-donor may increase the risk of thrombosis. Patients with cardiac insufficiency are at increased risk of thrombosis due to factors such as cardiac chamber enlargement, wall motion disorders, weakened contractility, and low cardiac output [11]. TTTS-donors often have cardiac insufficiency due to anemia and hypovolemia [12].

This study found no significant differences in thrombocytosis, TTTS recipients, and sepsis between the two groups. Studies suggest that the risk of thrombosis in secondary thrombocythemia is rare [13]. It has been reported that among 643 ICU patients with secondary thrombocythemia, only 1.5% developed venous thrombosis [14]. TTTS recipients may experience blood viscosity due to increased blood volume, thereby slowing blood flow and increasing the risk of thrombosis [15]. In patients with sepsis, the coagulation mechanism becomes vulnerable, resulting in activation of the coagulation system and subsequent thrombosis [16]. The results of this study do not support the above factors, although it is theoretically accepted that they are related to thrombosis, and catheter-related thrombosis will increase with PICC. Therefore, when indwelling PICC catheters are used in neonates with such high-risk factors, there is no need to worry about the risk of thrombosis, leading to more preventive anticoagulation measures.

Most of the thromboses associated with PICC catheterization occurred within 1 week after catheterization. The main clinical manifestations of thrombosis included edema, bruising or tenderness. Pain upon palpation of the affected limb suggests that the appearance of these clinical symptoms within 1 week following PICC insertion may be a warning sign for thrombosis, advising prompt vascular ultrasound confirmation. Previous studies have suggested that the axillary vein is more prone to PICC-related thrombosis. It is related to the anatomical position of the axillary vein. The diameter of the axillary vein is small, and it is located at the shoulder joint, which has a large range of motion. Repeated contact between the catheter and the vein wall can damage the integrity of the vascular endothelium and easily form thrombosis [17]. However, this study did not suggest a relationship between catheter placement and thrombosis. This may be attributable to the neonates enrolled in this study, who were premature infants. Additionally, it may be associated with the early post-natal catheter placement, with little voluntary activity and the risk of catheter position movement was very low. Some studies have found that when the catheter tip is not located in the lower one-third of the superior vena cava or is ectopic, the incidence of thrombosis is increased. When the catheter does not reach the superior vena cava, due to the small diameter of the venous lumen and the reduction of blood flow, it is easy to cause turbulence, prolong the contact between the medicinal solution and the intima, and increase the risk of endothelial injury [18]. Some of the cases in this study had the incorrect position of the tube end when the tube was placed. However, we did not find an increased risk of venous thrombosis in neonates with incorrect initial placement of the catheters. This was related to our use of bedside ultrasound to monitor the placement of the catheter. Upon discovery of incorrect tip positioning, timely interventions with minimal adverse impact were implemented to avoid catheter-related thrombosis, such as by pulling the catheter out by a few millimeters or reducing the concentration of the infusion liquid. This suggests that the dynamic monitoring of catheter position by bedside ultrasound in the NICU ward may significantly lower catheter-related thrombosis incidence.

Presently, there are few studies on the use of anticoagulant drugs after thrombosis in neonates, and there is no uniform recommendation for drug dose and duration. Low molecular weight heparin has become the anticoagulant of choice for primary and secondary prevention of venous thrombosis in many children [19]. The initial dose of low molecular weight heparin was 100 IU/kg for deep vein thrombosis and 50 IU/kg for superficial vein thrombosis, 1–2 times a day. Then, it was adjusted according to the disappearance of thrombus monitored by ultrasound. During the period, one case developed new intracranial hemorrhage during application of the drug; the drug was stopped immediately, and the hemorrhage was gradually absorbed in the later monitoring. It had no adverse prognosis. The mean duration of treatment was 10 days. Some studies recommend that the dose of unfractionated heparin should not exceed 75–100 IU/kg; the dose of low molecular weight heparin for anticoagulation therapy is 250–275 IU/kg/d [20]. The optimal duration of treatment should last between 6 weeks and 3 months. Some studies mentioned that during heparin anticoagulation therapy, the platelet count should be above 50 × 109/L, and the fibrinogen should be above 100 mg/dL [21]. For hospitals, in order to increase drug safety, it is best to monitor the anti-Xa activity of neonates using low molecular weight heparin. Blood is drawn 4 to 6 h after subcutaneous injection, and the target is 0.5 to 1.0 IU/mL; or tested 2 to 6 h after subcutaneous injection, with a target range of 0.5–0.8 IU/mL [18], so as to avoid the side effects of anticoagulation. In our study, when deep vein thrombosis was found, the catheter was removed after ultrasound showed that the thrombus disappeared, and the catheter was immediately removed when superficial vein thrombosis occurred. There is no unified recommendation for when to extubate the tube after catheter-related thrombosis. Some studies recommended that the PICC should be removed after 3 to 5 days of anticoagulant therapy [22].

This study had some limitations. The single-center, retrospective study design and low incidence of PICC-related venous thrombosis in the NICU may have introduced bias in patient selection and study results. Further research with large samples from multiple centers should be pursued to verify our findings.

In conclusion, neonatal PICC-related thrombosis is a serious complication of PICC catheterization. Severe cases can lead to loss of limb function. Our findings suggest that mothers with autoimmune diseases have an increased risk of PICC-related venous thrombosis in their neonates. PICC-related thrombosis should be considered for such neonates with other co-occurring high-risk factors for thrombosis. Furthermore, low molecular weight heparin calcium proved effective for treating PICC-related venous thrombosis. However, we should pay attention to monitoring side effects such as intracranial hemorrhage during the medication. If it occurs, most patients have a good prognosis if the medication is stopped in time.

## Figures and Tables

**Table 1 children-09-01126-t001:** Patient characteristics.

Groups	Number (*n*)	Gestational Age (Weeks)	Birthweight (g)	Male (%)
Thrombosis group	7	30.8 ± 1.9	1380 (1130, 1740)	71.4% (5/7)
Non-thrombotic group	28	30.8 ± 1.9	1295 (1122.5, 1500)	57.1% (16/28)
*p* value		0.858	0.412	0.690

**Table 2 children-09-01126-t002:** Univariate logistic regression analyses for PICC-related thrombosis.

Risk Factors	Thrombus	Non-Thrombus	*p* Value
Numbers	7	56	
Diabetes	42.9% (3/7)	21.4% (12/56)	0.342
Hypertension	42.9% (3/7)	37.5% (21/56)	1
Hypothyroidism	0% (0/7)	17.9% (10/56)	0.585
Autoimmune disease	28.6% (2/7)	1.8% (1/56)	0.03
Use of anticoagulants	42.9% (3/7)	7.1% (4/56)	0.025
Use of antihypertensive drugs	14.3% (1/7)	17.9% (10/56)	1
PICC beginning time (d)	5 (1, 6)	5 (3, 6)	0.521
The tip in the correct position	28.6% (2/7)	42.9% (24/56)	0.69
Left lower extremity vein	28.6% (2/7)	17.9% (10/56)	0.609
Right lower extremity vein	28.6% (2/7)	16.1% (9/56)	0.595
Left upper extremity vein	28.6% (2/7)	26.8% (15/56)	0.667
Right upper extremity vein	14.3% (1/7)	39.3% (22/56)	0.699
Wet lung	71.4% (5/7)	55.4% (31/56)	0.689
RDS	42.9% (3/7)	30.4% (17/56)	0.669
Aspiration syndrome	0% (0/7)	3.6% (2/56)	1
Intrauterine infectious pneumonia	0% (0/7)	21.4% (12/56)	0.329
PDA	28.6% (2/7)	55.4% (31/56)	0.243
Pulmonary hypertension	14.3% (1/7)	3.6% (2/56)	0.302
Cardiac insufficiency	14.3% (1/7)	1.8% (1/56)	0.211
TTTS-recipient	14.3% (1/7)	3.6% (2/56)	0.302
TTTS-donor	14.3% (1/7)	1.8% (1/56)	0.211
Erythrocytosis	0% (0/7)	1.8% (1/56)	1
Platelets before catheterization (×10^9^/L)	218 (96, 353)	217 (175, 270.25)	0.619
Thrombocytosis (≥500)	14.3% (1/7)	1.8% (1/56)	0.211
Thrombocytopenia	28.6% (2/7)	8.9% (5/56)	0.17
Feeding intolerance	28.6% (2/7)	17.9% (10/56)	0.609
Upper gastrointestinal bleeding	0% (0/7)	5.4% (3/56)	1
Asphyxia (mild to moderate)	14.3% (1/7)	16.1% (9/56)	1
Use of ibuprofen	0% (0/7)	26.8% (15/56)	0.182
Use of vasoactive drugs	28.6% (2/7)	12.5% (7/56)	0.26

Remarks: Vascular selection for PICC catheterization: the upper extremities include Guiyao vein, axillary vein, and cephalic vein, and the blood vessels of the lower extremities are the great saphenous vein.

**Table 3 children-09-01126-t003:** Multivariate logistic regression analyses for PICC-related thrombosis.

Variables	OR	*p* Value
Use of anticoagulants	10.632	0.043
Cardiac insufficiency	29.463	0.035
TTTS-donor	29.463	0.035
Autoimmune disease	14.239	0.090

**Table 4 children-09-01126-t004:** Clinical information on PICC-related thrombus.

Number	Vein Selection	Thrombus Location	Clinical Manifestations of Thrombosis	Platelet Count after Thrombosis (×10^9^/L)	Protein C (%)	Protein S (%)	AT-III (%)	Time from Catheter Placement to Thrombosis (d)	Treatment (Drug)	Low Molecular Weight Heparin	Course of Treatment (d)	How Long to Pull out the PICC after Thrombosis (d)	Thrombus Duration (d)	Prognosis
1	Left great saphenous vein	left hepatic portal vein	Systemic edema, lumbosacral and lower extremity edema	273	25	56	64	1	Nadroparin calcium injection	100 IU/kg, qd, 10 IU/kg, q12h, 12 d	12	12	The thrombus did not disappear until discharge; 20 days after birth, the thrombus disappeared during outpatient reexamination	Parents requested to be discharged
2	Left great saphenous vein	left external iliac vein	Edema at the PICC placement area	440	29	53	31	12	the same as above	100 IU/kg, q12h, 3 d; 75 IU/kg, q12h, 13 d	17	0.5 before thrombus	17	get well
3	Right basilic vein	right subclavian vein	Edema of right upper extremity, right head and neck, shoulder and back	57	26	70	43	36	the same as above	100 IU/kg, qd, 10 IU/kg, q12h, 4 d, 100 IU/kg, q12h, 2 d, 75 IU/kg, q12h, 3 d	10	2 before thrombus	5	get well
4	Right great saphenous vein	right popliteal vein	The lower limbs on the catheterization side were swollen and purple in color	273	17	21	22	6	the same as above	100 IU/kg, q12h, 3 d	3	2 before thrombus	3	get well
5	Left basilic vein	left upper extremity vein	The upper limbs on the catheterization side were swollen and purple in color	482	-	-	-	4	the same as above	100 IU/kg, q12h, 2 d (stopped due to intracranial hemorrhage)	2	6	5	get well
6	Right basilic vein	Right subclavian, axillary vein	The upper limbs on the catheterization side were swollen, purple, and the skin temperature increased	57	-	-	-	1	the same as above	100 IU/kg, q12h, 2.5 d; abnormal coagulation; changed to 10 IU/kg, q12h, 26 d	28	27	4	get well
7	Right great saphenous vein	right saphenous vein	The lower limbs on the catheterization side were red, swollen, with calluses and tenderness	254	17	47	37	5	the same as above	50 IU/kg, q12h, 3 d	3	0	3	get well

## Data Availability

The datasets used and analyzed during the current study are available from the corresponding author on reasonable request.

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
