# Peer review of "Clinical Characteristics of Venous Thrombosis Associated with Peripherally Inserted Central Venous Catheter in Premature Infants"

_children, 2022, doi:10.3390/children9081126_

Round 1

Reviewer 1 Report

Dear Authors!

I have one major and some minor remarks.

Major

You found two possible risk factors for venous thrombosis in a univariate analysis, which are mother’s autoimmune disease and use of anticoagulation (by mother, if I understand right). But, I see that thrombocytosis and thrombocytopenia could be also a potential risk factors. P-values are close to 0.2 which is more or less acceptable for a small sample. I recommend to perform multivariate analysis in attempt to confirm if mother’s autoimmune disease is a true independent risk factor. You can include in a calculation thrombocytosis and thrombocytopenia also.

Minor

Abstract

I would prefer to see a structured Abstract with Background, Materials, Results and Conclusions section

Materials and Methods

You mentioned total number of medical records that were analyzed to retrieve data on venous thrombosis in the Results section (3043).  But, it has to be mentioned in Materials in Methods also.

Statistical analysis: as I see you compared categorical variables also. Please, mention what test did you use.

Line 57. The statement “They were hospitalized during the same period” has to be removed as it repeats in line 59.

Line 61. Was based, not “is based”

Results

Details of anticoagulation in those who experienced vein thrombosis has to be place in this section. I mean what is written in lines 212-217.

As it was one serious adverse event while on anticoagulation (intracranial hemorrage) I recommend to add a subsection “Complications of anticoagulation”

Table 1. No need in a  row with Z/t/χ²

Table 2. No need in a column with Z/t/χ²

Lines 98-99. Please, reformulate this as to make it more understandable.

Line 105. What does “noble vein” mean?

Line 126. What does “thrombus duration” mean? Do you mean that thrombus resolved during this period? If so, then you should better you term “complete resolution”.

Discussion

Line 207. What does “blood routine” mean?

Lines 223-224. It seems you have meant that anti-Xa monitoring for those who receive anticoagulation?

Conclusions

You obtained data on rate of venous thrombosis and possible risk factor only. So, conclusions have to contain just this information.

Author Response

Answer to REVISER 1

Thanks for suggestions.

Major

You found two possible risk factors for venous thrombosis in a univariate analysis, which are mother’s autoimmune disease and use of anticoagulation (by mother, if I understand right). But, I see that thrombocytosis and thrombocytopenia could be also a potential risk factors. P-values are close to 0.2 which is more or less acceptable for a small sample. I recommend to perform multivariate analysis in attempt to confirm if mother’s autoimmune disease is a true independent risk factor. You can include in a calculation thrombocytosis and thrombocytopenia also.

Answer:

After the factors with P ≤ 0.2 were included in multivariate analysis,  P > 0.05, and the difference was not statistically significant.

Minor

Abstract

I would prefer to see a structured Abstract with Background, Materials, Results and Conclusions section

Answer:I have revised in the manuscript.

Materials and Methods

You mentioned total number of medical records that were analyzed to retrieve data on venous thrombosis in the Results section (3043).  But, it has to be mentioned in Materials in Methods also.

Answer:I have added the total number of PICC placements to the Materials and methods section.

Statistical analysis: as I see you compared categorical variables also. Please, mention what test did you use.

Answer:Categorical variables were assessed through the χ2 or the Fisher exact test, as appropriate.It is in lines 82 and 83. 

Line 57. The statement “They were hospitalized during the same period” has to be removed as it repeats in line 59.

Answer:I have removed this sentence.

Line 61. Was based, not “is based”

Answer:I have revised.

Results

Details of anticoagulation in those who experienced vein thrombosis has to be place in this section. I mean what is written in lines 212-217. 

Answer:I have revised.

As it was one serious adverse event while on anticoagulation (intracranial hemorrage) I recommend to add a subsection “Complications of anticoagulation”

Answer:I have added this section.

Table 1. No need in a  row with Z/t/χ²

Answer:I have deleted.

Table 2. No need in a column with Z/t/χ²

Answer:I have deleted.

Lines 98-99. Please, reformulate this as to make it more understandable. 

Answer:Vascular selection for PICC catheterization:the upper extremities include (Guiyao vein, axillary vein, and cephalic vein), and the blood vessels of the lower extremities are the great saphenous vein.

Line 105. What does “noble vein” mean?

Answer:Guiyao vein.

Line 126. What does “thrombus duration” mean? Do you mean that thrombus resolved during this period? If so, then you should better you term “complete resolution”.

Answer:Your understanding is right.I have revised.

Discussion

Line 207. What does “blood routine” mean?

Answer:It means routine blood test.

Lines 223-224. It seems you have meant that anti-Xa monitoring for those who receive anticoagulation?

Answer:Yes.If the hospital can do it, it can decrease the adverse side of anticoagulation.

Conclusions

You obtained data on rate of venous thrombosis and possible risk factor only. So, conclusions have to contain just this information. 

Answer:I have revised.

Answer to REVISER 2

Thanks for suggestions.

The importance of the findings of this study has been further emphasized in the conclusion.

Reviewer 2 Report

The study is interesting and may have important clinical implications. The authors aimed to analyze clinical characteristics and risk factors for peripherally inserted central catheter placement in premature infants. Even if the number of patients included in the study is very small, I consider that the results presented may have important clinical implications. I think that the study may be published after a few minor revisions. I consider that the authors should better emphasize the importance of the results obtained and also should address the future scope and topics that are important and that could not be covered in the manuscript. 

Author Response

Thanks for suggestions.

The importance of the findings of this study has been further emphasized in the conclusion.

Round 2

Reviewer 1 Report

Dear Authors!

Thank you for addressing my remarks. I'm Ok with the paper now

Author Response

Thanks very much for your advice.
